# Plant Growth Regulators Improve the Production of Volatile Organic Compounds in Two Rose Varieties

**DOI:** 10.3390/plants8020035

**Published:** 2019-01-31

**Authors:** Mohammed Ibrahim, Manjree Agarwal, Jeong Oh Yang, Muslim Abdulhussein, Xin Du, Giles Hardy, Yonglin Ren

**Affiliations:** 1School of Veterinary and Life Science, Murdoch University, 90 South St., Murdoch, WA 6150, Australia; m.ibrahim@murdoch.edu.au (M.I.); m.agarwal@murdoch.edu.au (M.A.); y.ren@murdoch.edu.au (Y.R.); 2Faculty of Agriculture, Al Qasim Green University, Babylon 51002, Iraq; 3Plant Quarantine Technology Centre, Animal and Plant Quarantine Agency (APQA), Gimcheon 39660, Korea; joyang12@korea.kr; 4Faculty of Agriculture, University of Kufa, Najaf 54003, Iraq; muslim.alrubaye@uokufa.edu.iq

**Keywords:** benzyladenine, naphthalene acetic acid, Hybrid Tea, Floribunda, HS-SPME, GC-MS

## Abstract

The study focused on the influence of the plant growth regulators (PGRs) benzyladenine (BA) and naphthalene acetic acid (NAA) on the production of volatile organic compounds (VOCs) from the flowers of two modern rose varieties, Hybrid Tea and Floribunda. Thirty-six plants of Hybrid Tea and Floribunda were tested. Benzyladenine and naphthalene acetic acid were applied at 0, 100 and 200 mg/L to both rose varieties. Gas chromatography, coupled with flame ionization detection and mass spectrometry, was used to analyze and identify the volatile organic compounds from the flowers. A three-phase fiber 50/30 µm divinylbenzene/carboxen/polydimethylsiloxane was used to capture VOCs, at 2, 4 and 8 weeks, and 4 weeks was selected as it had the highest peak area. In total, 81 and 76 VOCs were detected after treatment of both rose varieties with BA and NAA, respectively. In addition, 20 compounds, which had significant differences between different treatments, were identified from both rose varieties. The majority of VOCs were extracted after the application of 200 mg (BA and NAA) /L of formulation, and four important compounds, cis-muurola-4(141)5-diene, y-candinene, y-muurolene and prenyl acetate, increased significantly compared to the controls. These compounds are commercially important aroma chemicals. This study used the rapid and solvent-free SPME method to show that BA and NAA treatments can result in significant VOC production in the flowers of two rose varieties, enhancing the aromatic value of the flowers. This method has the potential to be applied to other valuable aromatic floricultural plant species.

## 1. Introduction

Most floral scents are considered pleasant; they have played key roles in the early development of flowers [1]. The genus *Rosa* is comprised of between 100 and 200 species and more than 18,000–24,000 commercial cultivars most of which are hybrids; collectively, these are based on only eight wild species [2,3,4]. Hybrid tea and Floribunda roses are considered modern roses and are popular in Australia [5]. The main commercial and industrial uses of roses are based on the perfume and floriculture industries, the former of which relies on scented rose varieties [6]. There are many studies on the emission of rose volatiles but few discuss modern rose varieties [7]. Many factors affect the emission of aromatic compounds, among which are plant growth regulators (PGRs) [8].

Plant growth regulators (PGRs) are produced naturally by plants or synthetically by chemists and play a key role in life cycles of plants [9]. PGRs include cytokinins and auxins, which are small molecular compounds that at low concentrations regulate plant developmental processes [10]. Benzyladenine (BA) and naphthalene acetic acid (NAA) have both been applied pre- and post-harvest to many ornamental plant species. Cytokinins such as BA are used to improve the quality and vase life of cut flowers. Moreover, BA promotes cell elongation and division, decreases flower drop and increases flower production [11]. BA has also been used to improve the growth and development of aromatic plants. Treating *Lantana camara* L. plants with BA at 0.44 and 4.4 µmol/L increased volatile organic compound (VOC) production, which was detected by solid phase micro-extraction (SPME) [12]. Similarly, Zielińska et al. [13] indicated that the production of VOCs increased with the application of different PGRs, such as 0.57 µM indole-3-acetic acid (IAA), 0.45 µM thidiazuron, 9.3 µM kinetin and 4.4 µM BA, when applied to *Agastache rugosa* (Fisch. Et C. A. Mey.) plants grown in vitro. Another study on *Stevia rebaudiana* L. plants reported that BA and IAA added to tissue culture media significantly influenced the in vitro production and distribution of secondary metabolites to produce strong antioxidant activity [14]. Auxins such as NAA coordinate many growth and behavioral processes in plant life cycles. Furthermore, NAA induces cell division and elongation, plays a key role in shoot development, and increases flower production [10]. A study on *Rosa damascena* (Damask rose) indicated that the application of NAA at 25 mg/L to rose plants, increased flower longevity, plant height, flower yield and flower oil content [15]. A study by Bota and Constantin [16] showed that the addition of 0.1 mg/L NAA and 1 mg/L BA to a cell suspension culture of *Digitalis lanata* (Grecian foxglove) increased the production of flavonoids, which are related to secondary metabolite production. Another study, by Coste et al. [17], indicated that the addition of BA (0.2 mg/L), kinetin (0.1 mg/L) and NAA (0.05 mg/L) together with a mineral salt (MS) medium enhanced the production of secondary metabolites in the shoot cultures of *Hypericum hirsutum* and *H. maculatum* plants. Little information is known about the influence of these PGRs on secondary metabolite production [18]. To our knowledge, there has been no research reported to date on the influence of NAA on volatile production in rose plants.

The detection of VOCs is widely used by scientists because of their wide range of beneficial pharmacological and biological activities [19]. Currently, more than 400 VOCs have been identified from rose flowers [20]. The headspace solid phase micro-extraction (HS-SPME) is a method for the extraction of VOCs from plants; it is now commonly used [21]. HS-SPME is a pre-concentration technology, which integrates extraction, concentration and sampling, and is a simple, fast and solvent-free preparation method for the extraction of VOCs [22]. It also prevents the production of artifacts compared with conventional solvent extraction procedures [23]. This method, coupled with gas chromatograph GC-FID/MS analysis and using the SPME fiber divinylbenzene/carboxen/polydimethylsiloxane (DVB/CAR/PDMS), has been successfully applied in the analysis of VOCs emitted from rose plants [24,25]. To our knowledge, there are no studies on the influence of BA and NAA on volatile production from rose flowers using the SPME technique. Consequently, the aim of this study was to study metabolite accumulation in the flowers of two rose varieties following the application of BA and NAA, using the HS-SPME technique with GC-FID/MS.

## 2. Materials and Methods

### 2.1. Plant Material and Maintenance

The experiment was conducted in March 2018 (summer). Thirty-six plants of two rose varieties (18 plants each of Hybrid Tea cv. ‘Mr. Lincoln’ and Floribunda cv. ‘Iceberg’) were grown in an evaporatively cooled greenhouse at Murdoch University (Perth, Western Australia, Australia). Next, 100 and 200 mg of BA and NAA were mixed with 4 to 6 drops of liquid sodium hydroxide (NaOH) 1 N and then added to 1 L of double distilled water. The growing conditions, age of plants, application of BA and NAA, and plant maintenance were described previously [25]. The rose plants were prepared in a factorial-based and completely randomized design with two rose varieties, two hormones, three concentrations of each hormone and three replicate plants for each treatment. The experiment was repeated once.

### 2.2. Sample Extraction Using HS-SPME

The rose plants were transferred to the laboratory for sampling. The samples were analyzed in biological triplicates. Three undamaged and fully open flowers for each rose variety were individually sampled within a chamber for the extraction of the VOCs, as previously described [24]. Briefly, the flowers were placed in a 500 mL glass jar with a septum and sealed with aluminum foil to trap the volatiles inside the glass jar. The SPME fiber was inserted through a 5 mm port within the septum to absorb the VOCs. The HS-SPME protocol included three steps: First, the SPME fiber was activated for half an hour in the GC-FID; second, the extraction time for the samples was 2 h with the HS-SPME; and, finally, the SPME fiber was injected into the GC port for 45 min at 250 °C to collect the VOCs from the flowers of the two rose varieties.

### 2.3. Standards and Reagents

The n-alkane standard (C7-C30) reference material at 1000 μg/mL in hexane was purchased from Sigma-Aldrich (catalog number 49451-U; Castle Hill, NSW, Australia). Naphthalene acetic acid (NAA) (95% purity, product number N0640) was purchased in crystalline form from Sigma-Aldrich (Castle Hill, NSW, Australia). Mohammed et al. [25] described all other standards and regents. Briefly, these included high-performance liquid chromatography grade ethanol, n-hexane (95%), benzyladenine (BA) (98%), sodium hydroxide as a pellet (97%), rooting hormone (Clonex) in gel form, vapor guard (anti-transpirant concentrate) liquid, deionized water (Milli-Q Ultrapure water system) and double distilled water.

### 2.4. Apparatus and Equipment

A semi-quantitative analysis of VOCs from the flowers of Hybrid Tea and Floribuna rose varieties was performed to obtain the relative amount of each volatile component. For this procedure, a gas chromatograph with a flame ionization detection (GC-FID) system, 7829A GC (serial number CN14272038; Mulgrave, Victoria, Australia), equipped with an HP-5MS non-polar column (30 m × 0.25 mm, film thickness 0.25 μm, catalogue number 13423) was used. The column temperature program was between 50 to 250 °C with gradual increases of 5 °C/min; helium was used as a carrier gas with a flow of 1.1 mL/min. The total run time was 45 min; the GC-FID mode was splitless and the temperatures of the injector were 250 °C and 290 °C, respectively.

The flower samples were analyzed and identified using a GC-MS Agilent 7820A gas chromatograph coupled with a 5977E mass spectrometer. The analytical conditions used were as follows: splitless injection at 250 °C, and an HP-5MS capillary column (30 m × 0.25 mm fused-silica, with a film thickness of 0.25 μm, Mulgrave, Victoria, Australia). The column temperature program was 50 to 250 °C with gradual increases of 5 °C/min; helium was used as a carrier gas with a flow of 0.7 mL/min. The total GC-MS run time was 45 min and the constituents were identified by matching their spectra with those recorded in a mass spectra library (Wiley, Hoboken, NJ, USA). Relative amounts were calculated in relation to the total area of the chromatogram. Mass spectra were compared with the retention indices (RI) or GC retention time data of synthetic standards and compounds published in the literature. The same conditions of the GC-MS analysis were employed to ensure reproducibility by using the RESTEK website to translate the conditions (the pure chromatography EZGC method translator) of the GC-FID to GC-MS. A 50/30 µm divinylbenzene/ carboxen/polydimethyl siloxane (DVB/CAR/PDMS) coated fused-silica fiber was used as it is the most suitable fiber for adsorbing VOCs emitted from the headspace of different rose tissues [24,25].

### 2.5. Optimization of Different Sample Times after PGR Application

To determine the most efficient collection time for VOCs after the application of BA and NAA, samples were collected at 2, 4 and 8 weeks post-PGR treatment.

### 2.6. Assessment of Peaks and Identification of Volatile Compounds

The volatile compounds were identified using quantitative analysis software (MS quantitative analysis) for GC identification. Each treatment was compared with the other treatments according to the total peak area. The volatile compounds identified from samples were also compared with the compounds identified from the internal standards. Moreover, the identified volatile compounds were also compared with the literature using the retention index (RI) relative to a mixture of n-alkanes (C7–C30) in n-hexane by using DVB-CAR-PDMS fiber. Comparison of the MS-fragmentation pattern of the target analyses with those of pure components was also performed. The VOC peaks produced by the GC-MS were identified using AMDIS [25].

### 2.7. Statistical Analyses

Analyses of variances (ANOVA) were made using the general liner model procedure of the statistical analysis software (SAS®) University edition. The means of 10 main peak areas were analyzed at each of the sampling times. The least significant differences (LSD) and principal component analysis (PCA) with one way ANOVA were analyzed with the online program MetaboAnalyst 4.0 (http://www.metaboanalyst.ca/faces/home.xhtml) to compare the means, the least significant differences (LSD) are reported at *p* ≤ 0.05 [25]. The principal component analysis explains the differences between the various PGR concentrations, by means of factors obtained from the datasets, to determine which variables contribute the most to the differences. PCA was carried out to investigate whether two groups can be separated and used to determine their metabolic distinction. In addition, PCA was used to find relatively homogeneous clusters of observations based on the analyses and to identify VOCs. Score plots were also created to visualize the relationships among variables and to indicate likely discriminant variables.

## 3. Results and Discussion

### 3.1. Comparison of Optimal Sampling Times after PGR Application

The different sampling times (2, 4 and 8 weeks) were significantly (*p* < 0.05) different from each other for both the BA and NAA treatments. The peak areas of VOCs increased from 2 to 4 weeks and decreased at 8 weeks (Figure 1), so 4 weeks was selected as the optimum sampling time for both BA and NAA. This sampling time corresponds with the results of References [25,26] on rose plants. In contrast, a study by Sardoei et al. [27] indicated that samples collected from *Aloe barbadesis* plants at 8 weeks after the application of PGRs was the optimum time for sampling.

### 3.2. Analysis of VOCs after BA and NAA Application

A total of 81 components were identified from both rose varieties after the application of BA. Eight VOCs produced by Hybrid Tea flowers were significantly (*p* < 0.05) different from the controls (double distilled water) as determined by one-way ANOVA and post-hoc tests (Figure 2a). In contrast, the Floribunda flowers produced five VOCs that were significantly (*p* < 0.05) different from the controls (Figure 3a). The accurate collection of VOCs is very important for the analysis of flower volatiles. Various methods have been used to analyze volatile compounds, including HS-SPME [28]. Unfortunately, the VOCs obtained from extracted oils rarely provide the natural aroma of flowers because of thermal artifacts produced during the steam distillation process that typically occur at 60–70 °C [29]. The HS-SPME fiber method is favorable for the analysis of volatiles emitted from flowers because high-temperature artefacts are minimized during the process [30]. Overall, the different rose varieties showed differences in the quantity and quality of the volatile compounds produced (Table 1 and Table 2).

A total of 76 compounds were identified from the two rose varieties after NAA application. Five compounds showed significant (*p* < 0.05) differences for Hybrid Tea (Figure 2b), while two compounds were significant (*p* < 0.05) for Floribunda (Figure 3b), compared to the control. The two PGRs induced different types, amounts, and quantities of VOCs emitted from the flowers. The HS-SPME technique was used to analyze the emission of volatile organic compounds from fresh flowers of two rose varieties [24] and chrysanthemum [31]. The results correspond to a recent study by Ibrahim et al. [25], which showed that as the concentration of BA applied to Hybrid Tea and Floribunda increased from 0 to 200 mg/L, the number of metabolites produced increased significantly.

### 3.3. Influence of BA and NAA on VOCs Emitted from Flowers of Two Rose Varieties

For both BA and NAA, the application of 200 mg/L gave the highest peaks compared to 0 or 100 mg/L for both rose varieties (Figure 4). Similarly, Ibrahim et al. [25] showed that BA at 200 mg/L increased the peak areas and the emission of volatile compounds from different rose tissues in two rose varieties (Hybrid Tea and Floribunda). Farooqi et al. [32] indicated that the amount of essential oil extracted from *Rosa damascena* plants after the application of kinetin increased compared to the control plants. Moreover, another study on *Plectranthus ornatus* plants grown in tissue culture indicated that BA application significantly increased the amount of VOCs and that these helped callus induction [33]. Prior to the current study, no work had been done on how NAA effects the production of VOCs in vivo in roses.

### 3.4. Identification of VOCs Emitted from Flowers of Two Rose Varieties

A diverse range of VOCs were produced from the flowers, as determined by principal component analysis (PCA). These are shown by the distinct clustering of rose samples according to BA treatments (Figure 5a,b) and NAA treatments (Figure 6a,b) for both rose varieties. Different VOCs were produced in response to the different BA and NAA concentrations. There was no overlap and a good separation in the VOCs was produced between the three concentrations of BA and NAA. Each PGR had different numbers of identifying compounds for the two rose varieties (Table 1 and Table 2). The VOCs increased with increasing levels of BA, for instance, cis-muurola-4(14), 5-diene, which is an important compound in the perfume industry, increased from 7.61 to 33.45 in total peak area as BA levels increased from 0 to 200 mg/L in Hybrid Tea flowers (Table 2). Ma et al. [34] showed that this compound was found in different tissues of *Artemisia annua* (Sweet Annie) plants. Another aroma chemical, γ-cadinene, which is used for essential oils and medicinal purposes [35], increased from 0.13 to 4.14 with an increase in BA from 0 to 200 mg/L in Hybrid Tea. Similarly, γ-muurolene increased from 9.24 to 29.91 as BA increased from 0 to 200 mg/L in Floribunda (Table 1). Folashade and Omoregie [36] demonstrated that γ-muurolene is an important compound found in *Lippia multiflora* and used in the essential oils industry. Moreover, a study on tissue cultured *Plectranthus ornatus* (Coleus) plants using BA and NAA showed that many compounds such as cis-Muurola-4(14),5-diene, γ-cadinene and γ-muurolene increased significantly and stimulated callus induction and plant growth [33]. In the current study, after NAA application at 200 mg/L, the amount of myrcene increased in Hybrid Tea. This compound is very important as an essential oil, flavor, and fragrance, as well as for health and medicinal purposes, and corresponds to the findings by Joshua et al. [37] from *Cannabis sativa* plants. Prenyl acetate was produced in high quantities in Hybrid Tea after NAA application at 200 mg/L and is commonly used for many commercial and chemical purposes. In addition, PGRs have a key role in plant life cycles, as well as having an influence on all physiological processes that occur in plants. Moreover, they influence the biochemistry and metabolic processes in plants [38].

In this study, we only selected 81 and 76 VOCs produced from BA and NAA, respectively, for further analysis. These were analyzed by MetaboAnalyst 4.0, which showed that only 20 VOCs increased significantly compared to the controls. None of the BA and NAA applications significantly decreased the concentrations of VOCs below the controls.

## 4. Conclusions

The best sampling time was 4 weeks after application of the two PGRs. Eighty-one and 76 VOCs were obtained from flowers from both varieties after the application of BA and NAA, respectively. Of these, twenty VOCs were produced at significantly higher concentrations than the controls. The application of both NAA and BA at 200 mg/L was the optimum concentration and superior to the control and 100 mg/L treatments. Plant growth regulators play an important role in regulating plant responses, such as cell division, cell elongation and plant physiological and biochemical processes. To our knowledge, this is the first study on BA and NAA to improve the production of aromatic volatile compounds emitted from fresh rose flowers. These findings will benefit industries where there is an interest in increasing the quantity and quality of volatile compounds from plant products. In addition, four VOCs known to be precursors of fragrant compounds were increased significantly after BA and NAA treatments, these were cis-muurola-4(14),5-diene, γ-cadinene, prenyl acetate and γ-muurolene. All of these compounds play important roles in aromatic plants. Consequently, BA and NAA at 200 mg/L can be used to increase and improve the quantity and quality of VOCs produced from roses and potentially other floricultural plants. Future research could explore using concentrations higher than 200 mg/L for both BA and NAA. Furthermore, future research could look at applying both BA and NAA together; this has not been done before. Together they might enhance the quality and abundance of VOCs produced.

## Figures and Tables

**Figure 1 plants-08-00035-f001:**
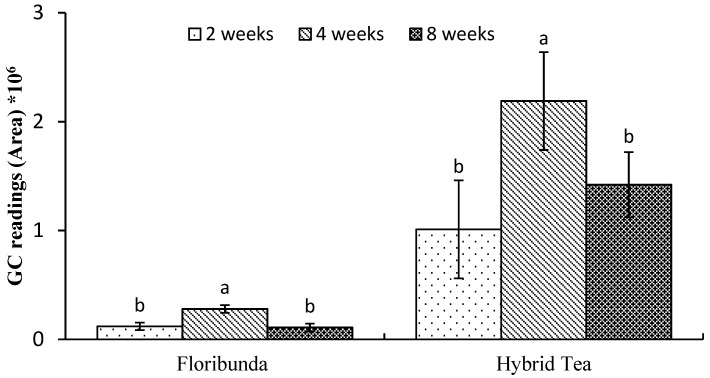
Comparison of the main peak areas at different sampling times for the analysis of volatile compounds collected from flowers of two rose varieties. Bars represent least significant differences (LSD) of the means at (*p* < 0.05) (*n* = 3).

**Figure 2 plants-08-00035-f002:**
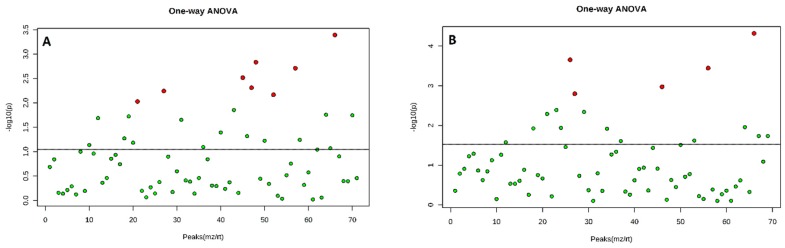
Volatile organic compounds collected from Hybrid Tea flowers after application of benzyladenine BA (**A**) and naphthalene acetic acid NAA (**B**), the red dots represent the volatile compounds which are significantly (*p* < 0.05) different to the total collected volatile compounds. Principal component analysis PCA analyses were used with one-way ANOVA.

**Figure 3 plants-08-00035-f003:**
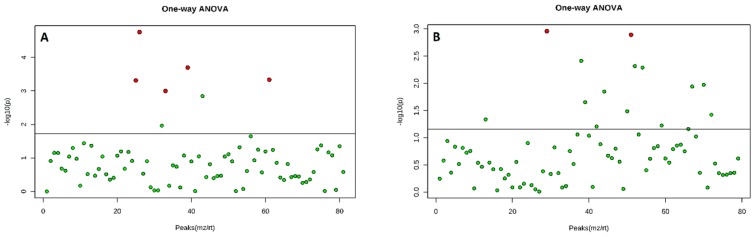
Volatile organic compounds collected from Floribunda flowers after the application of BA (**A**) and NAA (**B**), the red dots represent the volatile compounds which were significantly (*p* < 0.05) different to the total collected volatile compounds. PCA analyses were used with one-way ANOVA.

**Figure 4 plants-08-00035-f004:**
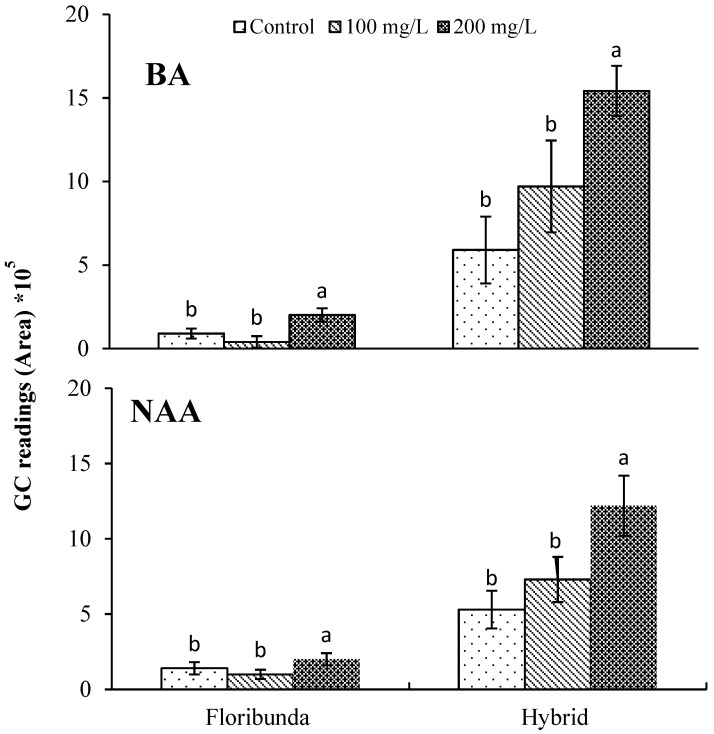
Effects of BA and NAA application on the extraction efficacy of the total VOCs identified from flowers of two rose varieties as determined by GC-MS. Error bars are LSD at (*p* < 0.05) (*n* = 3). One-way ANOVA analyses were used.

**Figure 5 plants-08-00035-f005:**
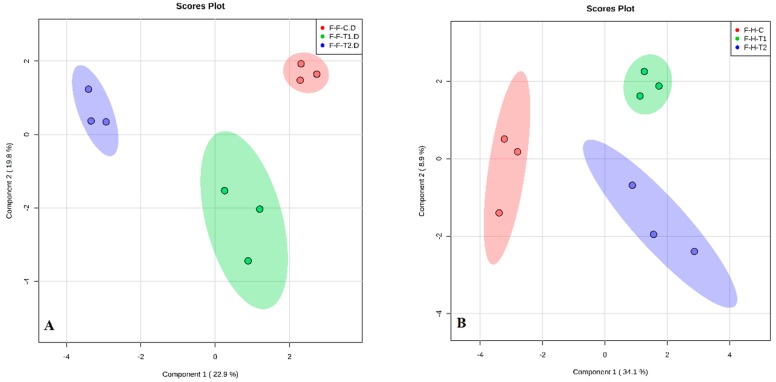
Principal component analysis (PCA) model obtained from the classification of volatile organic compounds emitted from the flowers of Floribunda (**A**) and Hybrid Tea (**B**), based on VOCs according to BA treatments. F and H represent Floribunda and Hybrid Tea, respectively. While C (control) (red), T1 (100 mg/L) (green) and T2 (200 mg/L) (blue) represent 0, 100 and 200 mg/L BA, respectively. Three dots in each group represent *n* = 3 biological replicates.

**Figure 6 plants-08-00035-f006:**
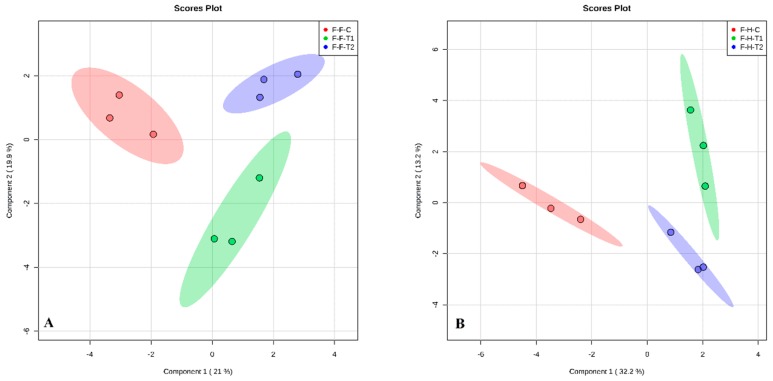
Principal component analysis (PCA) model obtained from the classification of volatile organic compounds emitted from the flowers of Floribunda (**A**) and Hybrid Tea (**B**), based on VOCs according to NAA treatments. F and H represent Floribunda and Hybrid Tea, respectively. While C (control) (red), T1 (100 mg/L) (green) and T2 (200 mg/L) (blue) represent 0, 100 and 200 mg/L NAA, respectively. Three dots in each group represent *n* = 3 biological replicates.

**Table 1 plants-08-00035-t001:** Identification of VOCs that differed significantly (*p* ≤ 0.05) between flowers of Floribunda rose after being treated with different concentrations of BA and NAA.

RT.	Compounds	RI	Mean of Peaks at BA (mg/L)			Mean of Peaks at NAA (mg/L)		
			0	100	200	0	100	200
8.82	Propyl glycolate	928	1.33	1.23	2.64 *	0.32	0.30	0.58
10.55	α-Pinene	939	3.56	4.02	6.40 *	4.61	5.60	5.43
13.52	D-Limonene	1018	0.11	0.84	1.09 *	2.52	2.03	2.38
25.24	Caryophyllene	1396	21.56	27.90	24.88	25.38	31.83	51.21 *
25.42	Isocaryophyllene	1424	27.46	28.72	22.86	17.03	34.62	37.11 *
26.59	γ-Muurolene ^†^	1494	9.24	12.06	29.91 *	26.66	23.97	27.09

(*) represents significant differences between different concentrations of BA and NAA (0 (control), 100 and 200 mg/L); RT—retention time; RI—retention index based on alkane series; ND—not detected; ^†^—compounds that increased significantly with LSD *p* ≤ 0.05 in Floribunda rose with increasing concentrations of BA and NAA compared to the control.

**Table 2 plants-08-00035-t002:** Identification of VOCs that differed significantly (*p* ≤ 0.05) between flowers of Hybrid Tea after being treated with different concentrations of BA and NAA.

RT.	Compounds	RI	Mean of Peaks at BA (mg/L)			Mean of Peaks at NAA (mg/L)		
			0	100	200	0	100	200
10.11	Prenyl acetate ^†^	932	2.28	2.09	2.69	1.98	12.95	17.01 *
12.77	Myrcene	979	12.25	16.60	13.20	14.12	17.48	30.44 *
19.19	Cuminal	1214	1.24	1.88	4.38 *	1.52	1.84	1.04
21.29	Geranial	1249	37.57	51.06	70.71 *	ND	ND	ND
22.03	Methyl geranate	1299	2.26	3.25	5.21 *	4.46	2.23	6.86
24.19	Geranyl acetate	1360	2.58	10.84	11.09 *	9.01	12.42	15.01
25.44	Caryophyllene	1428	2.26	3.25	5.21	4.04	6.18	7.39 *
25.61	α,β-Dihydro-β-ionone	1437	45.47	49.27	47.49	42.86	69.98	71.17 *
25.62	Dihydro-β-ionone	1439	44.82	56.10	76.17 *	25.22	24.75	38.47
25.88	cis-Muurola-4(14),5-diene ^†^	1455	7.61	10.83	33.45 *	5.07	10.43	13.17
30.90	trans-3-Octadecene	1695	1.07	1.26	2.20 *	4.28	6.99	7.04
34.27	Z-5-Nonadecene	1818	0.87	1.03	1.16 *	13.07	13.17	19.09
35.15	9-Nonadecene	1880	16.39	15.82	15.93	13.57	39.87	43.62 *
35.23	γ-Cadinene ^†^	1905	0.13	0.16	4.13 *	1.27	0.79	2.06

(*) represents significant differences between different concentrations of BA and NAA (0 (control), 100 and 200 mg/L); RT—retention time; RI—retention index based on alkane series; ND—not detected; ^†^—compounds that increased significantly with LSD *p* ≤ 0.05 in Hybrid Tea rose with increasing concentrations of BA and NAA compared to the control.

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
