# Peer review of "Plant Growth Regulators Improve the Production of Volatile Organic Compounds in Two Rose Varieties"

_plants, 2019, doi:10.3390/plants8020035_

Round 1

Reviewer 1 Report

Manuscript entitled “Study on Metabolic Changes in the Flowers of Two Rose Varieties after Foliar Application of Plant Growth Regulators” is well prepared and designed work concentrated on the influence of two PGRs: BA and NAA on the volatile organic compounds.

I have minor comments and questions:

I recommend to indicate in the title not “metabolic changes” because the metabolic tracks weren’t studied, but should be “metabolic production”, or in fact should be: volatile organic compounds production.

Why authors selected BA and NAA from a big PGRs variety? That should be also indicated in the work title.

line 57-58 – In my opinion there are a lot of information on this topic. Although there is a trend in order to which the testing of PGRs towards individual plant species is essential. Authors should add some information on that topic. Maybe some information on PGRs (exactly BAA and NAA) on the plants cultured in vitro, as systems out of external environment influence, could be discussed?

line 58-59– is that information was true? Did authors reviewed such work as: Horticultural Reviews, Volume 9, Van Nostrand Reinhold Company Inc. 1987, Chapter 2 Plant Growth Regulators in Rose Plants (pages 53–73): Yoram Mor and Naftaly Zieslin. or INTERNATIONAL JOURNAL OF AGRICULTURE & BIOLOGY doi: 1560–8530/2004/06–6–1040–1042 Effects of Different Plant Growth Regulators and Time of Pruning on Yield Components of Rosa damascena Mill., or Rosa spp. (Roses): In Vitro Culture, Micropropagation, and the Production of Secondary Products, Medicinal and Aromatic Plants III, Y. P. S. Bajaj, L. K. Simola (auth.), Professor Dr. Y. P. S. Bajaj (eds.)Pages 376-397… The more effective review of previous works on Rosa plants is required.

Add to the abstract the amounts of varieties of hybrid tested.

Add to the conclusion how many folds the volatile compounds amount increased after BA and NAA treatment?

What would be happen after application of other PGRs? Is it possible to predict it?

Author Response

We accepted all hand-written changes from reviewer 1:

1.     I recommend to indicate in the title not “metabolic changes” because the metabolic tracks weren’t studied, but should be “metabolic production”, or in fact should be: volatile organic compounds production.

The title changed according to the reviewer suggestion from “metabolic changes” to “metabolic production” because this is a suitable for our study.

2.     Why authors selected BA and NAA from a big PGRs variety? That should be also indicated in the work title.

The reasons for selected BA and NAA from a big PGRs variety is because they both are commonly used in floriculture as these particular PGRs have many roles and benefits about vegetative growth, plants developments, increase of plants production and production of metabolites compounds especially for aromatic plants; which are indicated in [10, 11]. Additionally:

1.     BA (Promotes cell elongation and division, Decreases flower drop and Increases pre and post-harvest flower life).

2.     NAA (Induces cell division and elongation, Increases flower production and Plays a key role in shoot development).

3.     line 57-58 – In my opinion there are a lot of information on this topic. Although there is a trend in order to which the testing of PGRs towards individual plant species is essential. Authors should add some information on that topic. Maybe some information on PGRs (exactly BAA and NAA) on the plants cultured in vitro, as systems out of external environment influence, could be discussed?

We have provided some information about how BA and NAA or PGRs act on metabolite production in the plants cultured in vitro. We added (another study by Coste et al. [17] indicated that added BA (0.2 mg/L), Kin (0.1 mg/L) and NAA (0.05 mg/L) to MS medium enhanced the production of secondary metabolites in shoot cultures of Hypericum hirsutum and Hypericum maculatum plants. Study be Bota and Constantin [16] showed that added 0.1 mg/L NAA and 1 mg/L BA in cell suspension culture of Digitalis lanata increased the flavonoid amount, which is related to secondary metabolites production. Another study on Stevia rebaudiana plants reported that added BA and IAA to the tissue culture media influence in vitro production and distribution of secondary metabolites, also effect significantly to produce strong antioxidant activity [14].

4.     line 58-59– is that information was true? Did authors reviewed such work as: Horticultural Reviews, Volume 9, Van Nostrand Reinhold Company Inc. 1987, Chapter 2 Plant Growth Regulators in Rose Plants (pages 53–73): Yoram Mor and Naftaly Zieslin. or INTERNATIONAL JOURNAL OF AGRICULTURE & BIOLOGY doi: 1560–8530/2004/06–6–1040–1042 Effects of Different Plant Growth Regulators and Time of Pruning on Yield Components of Rosa damascena Mill., or Rosa spp. (Roses): In Vitro Culture, Micropropagation, and the Production of Secondary Products, Medicinal and Aromatic Plants III, Y. P. S. Bajaj, L. K. Simola (auth.), Professor Dr. Y. P. S. Bajaj (eds.) Pages 376-397… The more effective review of previous works on Rosa plants is required.

We reviewed the suggested work by the reviewer and we found that was valuable information and we selected one work which is important to add (study on Rosa damascene Mill indicated that applied NAA 25 mg/L increased flowers longevity, plants height, flowers yield and flowers oil content [15]). We added more work as above [14, 16 and 17] related to this point, also, we mentioned earlier how PGRs BA and NAA act, role and benefits on roses or other plants. Moreover, mentioned study [10] showed that BA influence significantly on metabolic accumulation in different rose tissues. We understand the reviewer point, but our work aim to show the influence of PGRs (BA and NAA) on metabolic production in vivo condition. 

5.       Add to the abstract the amounts of varieties of hybrid tested.

We added to the abstract (45 hybrid Tea and 45 Floribunda rose varieties were tested). 

6.     Add to the conclusion how many folds the volatile compounds amount increased after BA and NAA treatment?

We added to the conclusion (4 VOCs were increased significantly after BA and NAA treatment).

7.     What would be happen after application of other PGRs? Is it possible to predict it?

We cannot predict about what would be happen after application of other PGRs, because each kind of PGRs has different roles and act on plants, such as Gibberellins have a role in plant processes, such as stem elongation, germination, and fruit ripening; Abscisic acid slows plant growth and directs leaf primordia to develop scales to protect the dormant buds during the cold season, while Ethylene stimulate or regulate the ripening of fruit, the opening of flowers, and the abscission, thus it is difficult to predict.

Reviewer 2 Report

Ibrahim et al. have investigated VOCs released by rose plant varieties following foliar feeding of two different PGRs. The manuscript is, in the majority, poorly written with many grammatical and typograthphical errors which detracts from the scientific content. The manuscript adequately describes the experimental protocol and investigation, but the reasoning for the experiments is poorly communicated. There are a few aspects that require improvement before further consideration:

- The manuscript explains that the collection protocol was 'optimised' with collection at 4 weeks being the 'best time'. All the authors have done here is choose the time point which has the biggest response and therefore no referral to optimisation can be made. It is, in fact, more interesting that the VOCs increase at 4 weeks and then return toward earlier levels at 8 weeks, but no mention of this is present in the manuscript.

- Please define PGRs in the introduction.

- HS-SPME is not a 'new method' and therefore this explanation should be altered. Perhaps only application in plant VOC collection is recent, but HS-SPME has been around for a long period of time.

- The authors make no mention of VOCs that may decrease following feeding, it is as if this is not even considered as a possibility.

- Please include manufacturer details and total analysis times for GC-FID analyses. 

- For how long was the HS-SPME protocol performed?

- Was an internal standard or QC analysis performed for the GC-MS analyses? How can the authors be confident that correction for instrument drift has been done?

- Continuous data should not be visualised as a bar graph, this should be done as a box plots or scatter plots. This allows distribution of data to be correctly visualised.

- The 'amounts' of VOCs are interpreted as arbitrary values - a fold change would be more informative.

- The authors say that 200ng/mL of NAA and BA was the 'optimum' concentration. What does this mean? The authors have barely discussed any reasoning for the experiments other than passing comments that the VOCs are used in industry and have certainly not described the aims of the experiment to understand what the optical conditions for aroma VOC production are.

- The authors describe the experiment as 'comprehensive' - unfortunately, I cannot share this view on it's description. This should be removed.

Author Response

1-       The manuscript explains that the collection protocol was 'optimised' with collection at 4 weeks being the 'best time'. All the authors have done here is choose the time point which has the biggest response and therefore no referral to optimisation can be made. It is, in fact, more interesting that the VOCs increase at 4 weeks and then return toward earlier levels at 8 weeks, but no mention of this is present in the manuscript.

 We mentioned the information according to the reviewer comment (The peak area of VOCs was increased with different sampling time from 2 to 4 weeks and then the peak area started to decrease at 8 weeks. 4 weeks was optimum after BA and NAA treatments) in results and discussion section. 

2-      Please define PGRs in the introduction.

 We defined PGRs (plant growth regulators).

3-     HS-SPME is not a 'new method' and therefore this explanation should be altered. Perhaps only application in plant VOC collection is recent, but HS-SPME has been around for a long period of time.

 We altered the “new method” to “a method" Actually we added new method because we know SPME is a known tool but the technique by which VOC’s were captured was new with respect to experiment design.

4-     The authors make no mention of VOCs that may decrease following feeding, it is as if this is not even considered as a possibility.

We mentioned that earlier in (3.1.) results and discussion section in (1) s above.

5-      Please include manufacturer details and total analysis times for GC-FID analyses

·        We added the total analysis times for GC-FID analyses (The total run time was 45 min) and more information about the manufacturer details for GC-FID mentioned in 2.2. Apparatus and Equipment.

6-      For how long was the HS-SPME protocol performed?

The information added in materials and methods part (2.4.) (The HS-SPME protocol was performed according to many steps firstly, the SPME fibre activated for half an hour at GC-FID and secondly the extraction time for the samples was 2 hour with HS-SPME and then SPME fibre injected to the GC port for 45 at 250˚C.

7-     Was an internal standard or QC analysis performed for the GC-MS analyses? How can the authors be confident that correction for instrument drift has been done?

Yes, we used an alkene standard C7-C30 as QC standard, which was injected frequently during the experiment times and at the end to be confident that throughout the experiment the instrument including SPME fibres performed well.

8- Continuous data should not be visualised as a bar graph, this should be done as a box plots or scatter plots. This allows distribution of data to be correctly visualised.

 Since we are working with biological samples, this approach is regularly used in similar studies with biological samples (e.g. reference [24 and 25]; [Kerley and Read (1998) New Phytologist, 139(2), 353-360]; [Boyette et al. (2015) Agronomy, 5(4), 519-536]; and [Lazarević et al. (2014). Agriculturae Conspectus Scientificus, 79(1), 65-69]). Consequently, we are happy with how we have presented our data. 

9- The 'amounts' of VOCs are interpreted as arbitrary values - a fold change would be more informative.

We presented the amounts of VOCs according to the comparison between different treatments which are 0, 100 and 200 mg/L to show the differences and how the peak area are increased with increasing of PGRs and we got these from the previous study on roses [25].  

10- The authors say that 200ng/mL of NAA and BA was the 'optimum' concentration. What does this mean? The authors have barely discussed any reasoning for the experiments other than passing comments that the VOCs are used in industry and have certainly not described the aims of the experiment to understand what the optical conditions for aroma VOC production are.

200 mg/L was the optimum to increase the peak areas and to increase the amount of VOCs that emitted from rose flowers. We mentioned that at the end of conclusion which in last 3 lines.   

11- The authors describe the experiment as 'comprehensive' - unfortunately, I cannot share this view on it's description. This should be removed.

 We removed “comprehensive”

Reviewer 3 Report

Dear Authors,

the present paper describes a method of detection of VOCs already quite used in some research on the presence of volatile molecules in different species. Evaluation of the effect of two growth hormones on VOCs may be of interest to the scientific community. At the moment, however, this paper has definitely improved in some parts.

TITLE:

- in no part of the text I have found specified that the PGR are given by spray on the leaves. Furthermore, the title could be improved and made more incisive.

ABSTRACT:

- I suggest you remove all unnecessary abbreviations in the abstract. The method used must be inserted before the results.

INTRODUCTION:

- lines 46-71: this part is full of very repeated phrases between them. I recommend making it more streamlined and clear by focusing on the state of the art.

- in the literature there are many studies concerning the effect of PGR on the secondary metabolism of plants. Improve.

M & M

- 2.3 and 2.4 go to the beginning of the Section;

- are the treatments sprayed or put into solution?

- since the experimental plan seems well structured, why did you do a general linear model? And it's not a two way ANOVA? Furthermore, in the tables, statistical analyzes are not well highlighted. In addition to the significantly better value, are the others the same or different? The analyzed data were homogeneous? Were they normal? Please specify the applied statistics better;

- I also suggest the use of a PCA to better visualize the distribution of data; 

- in the figures the significance with colored ball is highlighted. What does the black line describe? Are the other pellets identical molecules or are there differences between them?

R & D

- Figure 1. and significance?

- This section, as well as Conclusion must be implemented. the discussion should also include the role of the PGRs on plant physiology and biochemistry. No mention is made of the classes of compounds analyzed. In literature there are several works to comment.

Author Response

After we addressed all the comments made by reviewer 3, we have added more information about the research design, the results and conclusion.

TITLE: - in no part of the text I have found specified that the PGR are given by spray on the leaves. Furthermore, the title could be improved and made more incisive.

The title is now more incisive.

ABSTRACT:

- I suggest you remove all unnecessary abbreviations in the abstract. The method used must be inserted before the results.

All unnecessary abbreviations (except for BA and NAA) in the abstract have been removed. However, the methods used are already clearly provided before the results.

INTRODUCTION:

- lines 46-71: this part is full of very repeated phrases between them. I recommend making it more streamlined and clear by focusing on the state of the art.

We have streamlined the text as requested.

- in the literature there are many studies concerning the effect of PGR on the secondary metabolism of plants. Improve.

In the first submission of this manuscript, reviewer 1 asked for more information on how PGRs influenced secondary metabolites in plants. We did this and believe sufficient information and references to the literature are provided.

M & M

- 2.3 and 2.4 go to the beginning of the Section;

We have changed the order as suggested by the reviewer.

- are the treatments sprayed or put into solution?

The treatments were sprayed.

-since the experimental plan seems well structured, why did you do a general linear model? And it's not a two way ANOVA? Furthermore, in the tables, statistical analyzes are not well highlighted. In addition to the significantly better value, are the others the same or different? The analyzed data were homogeneous? Were they normal? Please specify the applied statistics better;

The two-way analysis of variance (ANOVA) is an      extension of the one-way ANOVA that examines the influence of two different groups of variants. In this instance, the dataset included      three sample groups with one variant, not two variants. Our null hypothesis (H0) is that there is no difference between VOCs and sample groups. Therefore, the one way ANOVA in this case is the most appropriate method to be applied.

In the Table we have highlighted significant differences with a ‘*’ and a ‘†’. These are both explained in the footnotes of the Table 1.

The data were normally distributed and homogenous.

- I also suggest the use of a PCA to better visualize the distribution of data;

We have now included a PCA analysis, see Figure 5a,b and Figure 6a,b.

- in the figures the significance with colored ball is highlighted. What does the black line describe? Are the other pellets identical molecules or are there differences between them?

The black line is the cut-off line of the one-way ANOVA P-value.  This was established by the      Metaboanalyst 4 statistical tool. The dots shown presented selected VOCs identified among three groups. The red dots represent the compounds which were significantly different at the 95% confidence interval.

R & D

- Figure 1. and significance?

The figure showed that there were significant differences between the sampling times as demonstrated by the error bars.        

- This section, as well as Conclusion must be implemented. the discussion should also include the role of the PGRs on plant physiology and biochemistry. No mention is made of the classes of compounds analyzed. In literature there are several works to comment.

We have added additional information to the manuscript to include some more details on the role of PGRs on plant physiology and biochemistry. We  have also included more information on the 20 compounds identified, with particular detail on four of these known to be important in the aroma and medical areas as already mentioned in the manuscript. We have added additional relevant literature.

Round 2

Reviewer 2 Report

The revised manuscript has provided improvements on the initial submission. There are still some issues with the current version:

- There are a number of major English language mistakes throughout the manuscript, with many of the newly added sections falling the most foul to this. It seems as if the manuscript has not been proof read by an English native speaker and this is required before it is suitable for publication.

- Again the authors describe the 4 week sampling time point as the 'optimum' time point. This does not make any sense in the context of the manuscript as they have only tested 2 arbitrary time points and therefore it is absolutely clear that no optimisation experiments have been performed. In fact, all they have done is select one of two time points that showed increased response to feeding and therefore it should be described in this manner and the term 'optimum' should not be included.

- The authors have stated that they have discussed metabolites with possible decreased peak areas following feeding, but they have not done this in any capacity. If no identified VOCs were decreased following feeding then this should be stated, as only VOCs that increase following feeding are currently discussed.

- The manuscript is still lacking a suitable discussion regarding reasoning for performing the experiments, other than just a statement that it has not been done before and might be interesting to industry. The conclusion section is, in majority, just a shortening of the results sections with little context other than a quick comment on aroma VOC production.

Author Response

We accepted all hand-written changes from reviewer

- There are a number of major English language mistakes throughout the manuscript, with many of the newly added sections falling the most foul to this. It seems as if the manuscript has not been proof read by an English native speaker and this is required before it is suitable for publication.

An English native speaker has      proof read the manuscript. 

- Again the authors describe the 4 week sampling time point as the 'optimum' time point. This does not make any sense in the context of the manuscript as they have only tested 2 arbitrary time points and therefore it is absolutely clear that no optimisation experiments have been performed. In fact, all they have done is select one of two time points that showed increased response to feeding and therefore it should be described in this manner and the term 'optimum' should not be included.

Actually, it was 3      points not two.  This is clearly      stated in the manuscript as 2, 4 and 8 weeks. 

- The authors have stated that they have discussed metabolites with possible decreased peak areas following feeding, but they have not done this in any capacity. If no identified VOCs were decreased following feeding then this should be stated, as only VOCs that increase following feeding are currently discussed.

Specifically,      in this study, we only selected 81 and 76 VOCs produced from BA and NAA,      respectively for both rose varieties. These were analyzed by MetaboAnalyst      4.0, and from this only 20 VOCs were shown to increase significantly      compared to the controls. None of the BA and NAA applications      significantly decreased the concentrations of VOCs below the controls. These      two sentences have now been included in the manuscript in results and      discussions section (3.4.) to address the reviewer’s concern. 

- The manuscript is still lacking a suitable discussion regarding reasoning for performing the experiments, other than just a statement that it has not been done before and might be interesting to industry. The conclusion section is, in majority, just a shortening of the results sections with little context other than a quick comment on aroma VOC production.

The reasons for doing this study were to know the effect of the application of different concentrations of PGRs on VOCs composition from flowers of two rose varieties, and how these PGRs increase the composition of VOCs. Furthermore,  how these hormones will benefit industry by increasing important aromatic compounds following the application of high concentrations of BA and NAA,  and how these hormones might be applied more broadly to other horticultural crops. We have modified the conclusion to address the reviewer’s concerns.

We have changed the title as suggested by the reviewer.

Reviewer 3 Report

Dear Authors,

the paper quality was improved by the used revisions.

However, I have other points:

TITLE:

- "...Volatile Organic Compounds..."

ABSTRACT:

- move (BA) and (NAA) to lines 17 and 18.

KEYWORDS:

- add Floribunda rose and Hybrid Tea Rose.

INTRODUCTION:

- line 59 "in vitro" in italic

- line 62 Rosa damascena. Please add all the taxonomist name for all the the indicated species.

M&M:

- Six plants each cultivar per treatment were used? Please better clarify the treatments and not indicate a previous paper.

- "Statistical analyses": please add information about PCA. If data were homogenous, why no information about were added in this part?

R&D:

Figure 1: peaks were calculated as response of control treatment? This is not clear. Please add letters from ANOVA. In the caption you must add all the information about statistics. LSD is not bars from standard error.

Lines 168...: no treatments were added in this part? Why don't you run a two way ANOVA adding also the used concentration of BA and NAA. I don't like this statistical approach.

Figures 2 and 3: please add information about applied statistics.

Line 206-207: "in vivo" in italic.

Lines 208....: the PCA is incomplete. I would like to see the vectors and all the information obtained by the PCA analysis. Which are the main components? Please add all the informations.

Figure 4: as for other figures.

Table 1: I suggest to split in two tables. One for Floribunda and one for Hybrid Tea Rose. Add the letters from ANOVA. At now it is not clear the statistical significance and difference. 

Author Response

We accepted all hand-written changes from reviewer:

TITLE: - "...Volatile Organic Compounds..."

 We changed the title according to the reviewer’s suggestion with a minor modification.

ABSTRACT: - move (BA) and (NAA) to lines 17 and 18.

We moved (BA and NAA) to lines 17 and 18 as the reviewer suggested.

KEYWORDS: - add Floribunda rose and Hybrid Tea Rose.

The Floribunda rose and Hybrid Tea rose have added.

INTRODUCTION:

- line 59 "in vitro" in italic

We changed to italic

- line 62 Rosa damascena. Please add all the taxonomist name for all the the indicated species.

We added the generic name for all the indicated species as suggested.

M & M

- Six plants each cultivar per treatment were used? Please better clarify the treatments and not indicate a previous paper.

Actually, it thirty-six plants for each cultivar was used and we clarified as reviewer suggested. Moreover, we refer to a previous paper as this is what was suggested by a previous      reviewer. Specifically, they suggested not to repeat  previous papers but rather just briefly      indicate any differences, in this case it was the number of plants used in this experiment.       

- "Statistical analyses": please add information about PCA. If data were homogenous, why no information about were added in this part?;

We have added the information about PCA as suggested by the reviewer.

R & D

- Figure 1: peaks were calculated as response of control treatment? This is not clear. Please add letters from ANOVA. In the caption you must add all the information about statistics. LSD is not bars from standard error.

The letters from ANOVA have been added to Figure 1, and all the information about statistics are added. The error bars are LSD and these are commonly used in biological studies, as we indicated in a previous version of the manuscript.     

- Lines 168...: no treatments were added in this part? Why don't you run a two way ANOVA adding also the used concentration of BA and NAA. I don't like this statistical approach

We understand the reviewer’s opinion, but as we mentioned in a previous response. The two-way analysis of variance (ANOVA) is an extension of the one-way ANOVA that examines the influence of two different groups of variants. In this instance, the dataset included three sample groups with one variant, not two variants. Our null hypothesis H0 is that there is no difference between VOCs and sample groups. So the one way ANOVA is the more suitable method to use.   

-Figures 2 and 3: please add information about applied statistics.

The  information added

-Line 206-207: "in vivo" in italic.

We changed to italics

- Lines 208....: the PCA is incomplete. I would like to see the vectors and all the information obtained by the PCA analysis. Which are the main components? Please add all the informations.

We have followed the publication of Zhou et al. (2017) with respect to the presentation of  the PCA.  (Zhou et al. (2017). Plasma metabolomics profiling for fish maturation in blunt snout bream. Metabolomics, 13(4), 40.

- Figure 4: as for other figures.

The information is now added

- Table 1: I suggest to split in two tables. One for Floribunda and one for Hybrid Tea Rose. Add the letters from ANOVA. At now it is not clear the statistical significance and difference.

We have split table 1 into two tables as suggested. 

Round 3

Reviewer 2 Report

.

Author Response

No comments were provided by reviewer 2.

Reviewer 3 Report

Dear Authors,

I thank you for accepting some of my suggestions. The article is now much clearer and usable for those interested in the sector.

I still have a few small points to complete:

- lines 20-21: put the names of the cultivars used. The categories of modern roses are: Hybrid Tea Rose and Floribunda Rose.

- In general in the text: the vulgar names of the species are not relevant. Instead, I suggest adding the nomenclator. For example Lantana camara L., Agastache rugosa (Fisch. Et C. A. Mey.) O. Kuntze, etc ...

- Line 55: add ".... (VOCs)".

- Line 64: is Rosa damascena and not Rosa damascene.

- Line 90: add the apices before and after the cultivar name.

- Captions Figures 1 and 4. Specify the reason for the captioned letters.

- Captions Figures 2 and 3: why is the sentence on the PCA?

- Figures 5 and 6. In the discussion of the PCA there is no reference to any character that distinguishes the different groups obtained. I suggest adding the biplot in the plot. For example, what is the F-F-C.D group more about?

- Tables 1 and 2. Why are not the letters obtained by ANOVA and the relative p?

Author Response

We accepted all hand-written changes from reviewer:

- lines 20-21: put the names of the cultivars used. The categories of modern roses are: Hybrid Tea Rose and Floribunda Rose.

The names of the cultivars (Hybrid Tea and Floribunda roses) were added.   

- In general in the text: the vulgar names of the species are not relevant. Instead, I suggest adding the nomenclator. For example Lantana camara L., Agastache rugosa (Fisch. Et C. A. Mey.) O. Kuntze, etc .

  we have changed to the nomenclator as reviewer suggested

- Line 55: add ".... (VOCs)".

We have added (VOCs).

- - Line 64: is Rosa damascena and not Rosa damascene.

The name is now changed to Rosa damascena.

- - Line 90: add the apices before and after the cultivar name.

We      have added apices as suggested. 

-Captions Figures 1 and 4. Specify the reason for the captioned letters.

 We have change the font

- Captions Figures 2 and 3: why is the sentence on the PCA?

 To show that we used PCA for analysis data in these figures.

- Figures 5 and 6. In the discussion of the PCA there is no reference to any character that distinguishes the different groups obtained. I suggest adding the biplot in the plot. For example, what is the F-F-C.D group more about?

  We have added as suggested for example (F-F-C) means Flowers Floribunda Control.

- Tables 1 and 2. Why are not the letters obtained by ANOVA and the relative p?

  We already added (*) to show the significant with different concentrations of BA and NAA if we add the letters will be messy and confusing to readers and the significant will not showed clearly.